# Space Target Tracking with the HRRP Characteristic-Aided Filter via Space-Based Radar

**Shuyu Zheng, Libing Jiang, Qingwei Yang, Yingjian Zhao and Zhuang Wang ***

National Key Laboratory of Science and Technology on Automatic Target Recognition, College of Electronic Science and Technology, National University of Defense Technology (NUDT), Changsha 410073, China; zhengshuyu21@nudt.edu.cn (S.Z.); jianglibing@nudt.edu.cn (L.J.); yangqingwei18@nudt.edu.cn (Q.Y.); drztsp@163.com (Y.Z.)
* Correspondence: wangzhuang@nudt.edu.cn; Tel.: +86-139-7516-3062

**Abstract:** Approaching space target tracking is a typical and challenging mission in the space situational awareness (SSA) field. As the space-based radar is able to monitor the space targets of interest full-weather all-time, the space-based radar system is utilized in this paper. However, most multi-target tracking (MTT) filters in target tracking studies merely utilize the location or narrow measurements, and many potentially valuable electromagnetic scattering characteristics are missed, which leads to space target false tracking problems. The space-based radar transmits a wide-band signal, and the measured high-resolution range profile (HRRP) information is an effective characteristic for different target discrimination. Therefore, the HRRP characteristics of space targets are implemented into the update recursion of the MTT filter, which can be utilized to improve the tracking performance. Then, to predict the target HRRP sequence, the geometrical theory of diffraction (GTD) model is utilized. Additionally, a modified spatial spectrum method with a novel covariance matrix is designed to improve the scattering parameter estimation accuracy. Finally, an adapting threshold is devised for merging the Gaussian mixture (GM) components weights. The proposed threshold is on the basis of the proposed HRRP characteristic-aided probability hypothesis density (PHD) filter, and it can tackle the problem of space target discrimination. Simulation results validate the effectiveness and robustness of the proposed probability hypothesis density (HGI-PHD) filter aided by HRRP information and improved with GM weights.

**Keywords:** space-based radar; space target tracking; high-resolution range profile (HRRP); the geometrical theory of diffraction (GTD) model; adapting Gaussian mixture (GM) weights; merging threshold

## 1. Introduction

### 1.1. Background and Problem Statement

The radar system in this paper is the space-based radar, and the reasons for utilizing a space-based radar to detect or track space targets are analyzed as follows. Note that with the rapid growth of stealth technology and space activity, the existing ground-based equipment fails to monitor our space targets of interest all-weather full-time. Compared with ground-based equipment and space-based optical equipment, a space-based radar can provide better coverage, and it is not constrained by the weather or the curvature of the Earth. Moreover, it possesses a stronger early warning ability for high-speed space targets. Therefore, the space-based radar is utilized to track space targets in this paper, and some valuable problems are solved under space-based observing scenarios.

The problems that exist in the above-mentioned space-based observing scenarios are essential for perceiving the orbital state of the space targets of concern in real-time accurately, which is a challenging mission for space situational awareness (SSA) [1,2]. In general, the SSA surveillance tasks conclude the effective observation of our space targets

of interest in the surveillance region, i.e., how the space target orbits change with varying time in order to estimate the individual characteristics or states (such as their positions and velocities in the reference coordinates). It should be mentioned that the number of space targets will change as time varies, and it is not known as a priori knowledge; therefore, the SSA tasks can be regarded as classical multiple target tracking (MTT) problems.

However, MTT will encounter some specific and typical problems when faced with the SSA surveillance tasks. For example, the space-based radar can only cover a limited tracking range due to its relatively smaller radar transmitting power compared to the ground-based radar system. Furthermore, it is challenging to track space targets accurately when they approach each other or the distance between them is close. What is more, all target state components sometimes are not provided thoroughly in a space-based radar system, and these characteristics make it more difficult and complicated for MTT problems.

An appealing and effective way to solve the aforementioned problems is using the random finite set (RFS) theory and finite set statistics (FISST). It is known that FISST is an elegant Bayesian formulation for the description of MTT based on the RFS theory [3,4]. Efficient solutions such as the probability hypothesis density (PHD) [5,6] and cardinalized PHD (CPHD) [7–9] have emerged in the MTT region in recent years. The core goal for the aforementioned filters is to update the first-order multi-target moments of the posterior probability density and update the global posterior probability density rather than recurrence, which is able to reduce the computational complexity to a large degree. Moreover, with the aim of alleviating poor cardinality estimation performance accuracy of the standard PHD filter, the CPHD filter is proposed to ease the above tension. It should be noted that the sequential Monte Carlo (SMC) model [10] and the Gaussian mixtures (GM) model [11] are the two most typical implementations of the above-mentioned Bayesian-based filters.

Note that among the various MTT filters, the PHD filter and the CPHD filter have concise formulation expressions and a slightly small computational burden, which is feasible for engineering applications. However, due to the "spooky action" in the process of PHD filter recursion, some limits exist as we propagate these two filters' legacy PHD. We would like to mention that it is difficult to derive a theoretical mathematical expression for the posterior probability density. Therefore, to tackle this problem, the multi-target cardinality-balanced multi-Bernoulli (CBMeMer) filter [12] is developed by propagating the posterior PDF. But under high-clutter density scenarios, the tracking performance of the CBMeMer filter will deteriorate sharply. In recent years, the generalized labeled multi-Bernoulli (GLMB) filter [13] and its computational simplified form, the labeled multi-Bernoulli (LMB) filter [14], have been proposed sequentially. These two filters are able to distinguish various target trajectories more accurately. However, considering the moment approximate operation, it is hard for the LMB filter to effectively avoid missed detection.

The space-based radar system is able to transmit a wide-band signal, and, thus, it can provide targets additional electromagnetic information like the high-resolution range profile (HRRP). Motivated by the above, we designed a filter that can update the target HRRP pseudo-likelihood. In order to predict the HRRP sequence and employ it in update recursion, the geometrical theory of diffraction (GTD) model is utilized in this paper. Being a classical scattering center model, the GTD model is effective in describing the radar targets' electromagnetic characteristics at high frequencies. In recent years, the GTD scattering center model has been widely applied in lots of radar domains, such as automatic target recognition (ATR), radar cross-section (RCS) extrapolation and interpolation, radar target three-dimensional (3D) reconstruction, and so on. Once the GTD scattering center is reconstructed, we can obtain the targets' electromagnetic scattering data, and the HRRP sequence can be predicted as well. Therefore, it is vitally important to construct a precise GTD scattering model and estimate the scattering parameters accurately. Thus, many effective methods, such as estimating signal parameters via the rational invariant technique (ESPRIT) algorithm [15–18], the multiple signal classification (MUSIC) algorithm [19–22], and the matrix enhancement and matrix pencil (MEMP) algorithm [23–25], have been proposed to estimate the scattering parameters from the back-scattered electromagnetic data.

Based on the aforementioned background as well as to deeply analyze the scientific problems in the process of tracking space targets based on a space-based radar, we separately analyzed the specific problems in the observation scene of a space-based radar and provided specific solutions so as to effectively improve the tracking accuracy of space targets.

### 1.2. Problem Analysis and Contributions

It is worth stressing that some electromagnetic scattering characteristics remain in radar signals, while most MTT filters fail to utilize them properly. Therefore, some drawbacks existed during the tracking recursion as follows:

1.  Generally, most MTT filters merely utilize target information such as distance and orientation but fail to use some targets' additional characteristics that a wide-band radar can provide, for example, the HRRP information.
2.  It is difficult to track space targets accurately when they are closely spaced with each other. So how can we improve the tracking performance when space targets are in dense clutter environments (due to thermal noise in space-based platforms, space debris, or other space targets)?
3.  If the HRRP characteristics are utilized in the MTT tracking recursion process, then how can we predict the HRRP sequence, and how can we compute the HRRP pseudo-likelihood for different targets?
4.  A more accurate GTD model is important for providing precise electromagnetic information for tracking update procedures. However, due to the thermal noise in a space-based platform, the signal-to-noise ratio (SNR) will be lower than the ground-based radar, which will lead to inaccuracy in scattering center parameter estimation. Therefore, it is vitally important to design a novel scattering parameter estimation method that can perform well in low-SNR scenarios.

The standard PHD filters do not utilize the targets' potential electromagnetic information abundantly, and they will encounter the problem of poor tracking performance under low-SNR scenarios or dense clutter environments. To improve the aforementioned problems, a modified PHD filter with HRRP characteristics is proposed in this paper, and the main contributions are shown as follows.

1.  To fully utilize the electromagnetic scattering characteristics of space targets, HRRP information is utilized in the update recursion of our proposed MTT filter. The HRRP for various targets and clutter differs significantly from each other as tracking time varies; thus, if the HRRP information is fully utilized, the difference between different targets or clutter will be enlarged, which is beneficial for improving the tracking accuracy.
2.  To solve the aforementioned second problem, an adaptive threshold for merging the Gaussian components is designed in this paper. The designed adaptive threshold is relevant to the noise covariance, and it is able to improve the tracking performance of the MTT filter. Furthermore, the discrimination ability between closely spaced targets can be enhanced at the same time.
3.  In order to predict the HRRP sequence, the GTD scattering center model is constructed in this paper. Then, the similarity rate between different HRRP sequences is denoted to describe the HRRP pseudo-likelihood.
4.  To solve the fourth problem mentioned above with the aim of improving the GTD scattering parameter estimation accuracy in low-SNR scenarios, a modified 3D-ESPRIT algorithm with a novel correlation matrix is proposed in this paper. Furthermore, by squaring the total covariance matrix, the parameter estimation accuracy can be further improved.

The rest of this paper is organized as follows. Section 2 investigates a review of the PHD filter and the GTD scattering center model. Furthermore, to obtain the time-varying observing angle, the observation geometry transformation for a space-based radar is also given in Section 2. Section 3 presents our novel scattering parameter estimation method and the analytic implementation of our proposed PHD (HGI-PHD) filter aided by HRRP characteristics and

improved with Gaussian mixture components. The simulation and performance evaluation are illustrated in Section 4. Finally, conclusions are drawn in Section 5.

## 2. Technical Foundation

In this section, we first introduce the basic theory of PHD filtering, which provides a foundation for space target tracking. Then, to predict the HRRP sequence of space targets, the GTD scattering center model is demonstrated in this section. Finally, the observation geometry transformation for the space-based radar is derived to compute the time-varying observing angle.

### 2.1. PHD Filtering

In order to ease the computational intractability of the Bayesian multiple-target filter, the PHD filter is proposed by using the first-order moment of an RFS, which can be called the intensity function, and it is used to approximate the posterior multiple-target state density and propagate posterior intensity. The intensity of an RFS is defined as follows:

$$v_k(x|Z_{1:k}) = \int \delta_X(x) f(X|Z_{1:k}) \delta X \tag{1}$$

where $\delta_X(x) = \sum_{w \in X} \delta_w(x)$ and $\delta_w(x)$ represent the Dirac delta function. $X$ and $Z$ represent the states and measurements, respectively. Here, $Z_{1:k}$ is omitted from the intensity to notate it more simply. The integral of the PHD is not a constant, and the expected number of targets is expressed as

$$\int v_k(x)\mathrm{d}x = \lambda_k \tag{2}$$

The PHD filter approximates the density of multiple targets by using a Poisson process as follows:

$$f_{k|k}(X) = e^{-\lambda_k} \prod_{x \in X} \lambda_k v_k(x) \tag{3}$$

Then, the PHD prediction recursion is computed as

$$v_{k|k-1}(x) = \int p_{S,k}(\zeta) f(x|\zeta) v_{k-1}(\zeta)\mathrm{d}\zeta + \gamma_k(x) \tag{4}$$

where $v_{k|k-1}(x)$ represents the predicted PHD density of time $k$, $p_{S,k}(\zeta)$ is the target survival probability, $\gamma_k(x)$ is the target birth intensity function, and $f(x|\zeta)$ denotes the Markov transition density with state $x$.

Finally, the PHD update step can be summarized as

$$v_k(x) = \left[1 - p_{D,k}(x)\right] v_{k|k-1}(x) + \sum_{z \in \mathbf{Z}} \frac{p_{D,k}(x) g_z(z|x) v_{k|k-1}(x)}{\kappa(z) + \int p_{D,k}(\zeta) g_z(z|\zeta) v_{k|k-1}(\zeta)\mathrm{d}\zeta} \tag{5}$$

where at time $k$, $p_{D,k}(x)$ is the detection probability of the target, $g_z(z|x)$ is the measurement likelihood of the target, and $\kappa(z)$ is the clutter RFS intensity function.

### 2.2. GTD Scattering Center Model

As one of the typical scattering center models, the geometric theory of diffraction (GTD) model enables us to describe the electromagnetic characteristics of radar targets effectively. Therefore, in a high-frequency region, the GTD scattering center model can be expressed as [26]

$$\begin{aligned} E(f_m, \theta_n, \varphi_k) &= \sum_{i=1}^{I} A_i (\mathrm{j}\tfrac{f_m}{f})^{\alpha_i} \exp[-4\pi\mathrm{j}f_m(x_i\cos\theta\cos\varphi + y_i\sin\theta\cos\varphi + z_i\sin\varphi)/\mathrm{c}] + \omega(f_m, \theta_n, \varphi_k) \\ &= \sum_{i=1}^{I} A_i (\mathrm{j}\tfrac{f_0 + m\Delta f}{f})^{\alpha_i} \exp[-4\pi\mathrm{j}f_m(x_i\cos\theta\cos\varphi + y_i\sin\theta\cos\varphi + z_i\sin\varphi)/\mathrm{c}] + \omega(f_m, \theta_n, \varphi_k) \end{aligned} \tag{6}$$

where $E(f_m, \theta_n, \varphi_k)$ represents the back-scattering data of radar targets, $I$ denotes the total scattering centers, and $\{A_i, \alpha_i, x_i, y_i, z_i\}$ denote the scattering intensity, scattering type, transversal position parameter, and longitudinal position parameter and vertical position parameter of the $i$-th scattering center, respectively. $f_m = f_0 + m\Delta f$ represents the operating frequency, and $\{f_0, \Delta f, m\}$ represent the initial frequency, the frequency step, and the frequency index, respectively. In similarity, $\theta_n$ is equal to $\theta_0 + n\Delta\theta$; $\varphi_k$ is equal to $\varphi_0 + k\Delta\varphi$; $\theta_0$ and $\varphi_0$ are the initial azimuth angle and the initial elevation angle, respectively; and $n\Delta\theta$ and $k\Delta\varphi$ are the radar rotation angles, which are relatively small. $c = 3 \times 10^8$ m/s represents the electromagnetic wave propagation speed, and $\omega(f_m, \theta_n, \varphi_k)$ denotes the Gaussian white noise. The type scattering parameter $\alpha_i$ of typical scattering structures is given in [26].

Then, according to Ref. [27], the scattering center parameters $\alpha_i$, $x_i$, $y_i$, and $z_i$ can be estimated as

$$\alpha_i = \frac{(|P_{xi}| - 1)f_0}{\Delta f} \tag{7}$$

$$x_i = \frac{-\text{angle}(P_{xi}) \times c}{4\pi\Delta f_x} \tag{8}$$

$$y_i = \frac{-\text{angle}(P_{yi}) \times c}{4\pi\Delta f_y} \tag{9}$$

$$z_i = \frac{-\text{angle}(P_{zi}) \times c}{4\pi\Delta f_z} \tag{10}$$

where $P_{xi} = (1 + \alpha_i\frac{\Delta f_x}{f_{x0}})\exp(-4\pi\text{j}\Delta f_x x_i/c)$, $P_{yi} = \exp(-4\pi\text{j}\Delta f_y y_i/c)$, and $P_{zi} = \exp(-4\pi\text{j}\Delta f_z z_i/c)$.

Finally, the intensity parameters $\tilde{A}$ can be estimated by using the least square method as follows

$$\tilde{A} = (G^\text{H}G)^{-1}G^\text{H}E_\text{k} \tag{11}$$

where

$$G = [a_1, \ldots, a_I] \tag{12}$$

$$a_i = \begin{bmatrix} a_i(0,0,0), \ldots, a_i(M-1,0,0), a_i(0,1,0), \ldots, \\ a_i(M-1,1,0), \ldots, a_i(M-1,N-1,0), \\ a_i(0,0,1), \ldots, a_i(M-1,N-1,K-1) \end{bmatrix}^\text{T} \tag{13}$$

$$a_i(m,n,k) = \text{j}\left(\frac{f_m}{f_0}\right)^{\alpha_i}\exp[\frac{-4\pi\text{j}f_m}{c}(x_i\cos\theta_n\cos\varphi_k + y_i\sin\theta_n\cos\varphi_k + z_i\sin\varphi_k)] \tag{14}$$

$$E_k = \begin{bmatrix} E(f_0,\theta_0,\varphi_0), \ldots, E(f_{M-1},\theta_0,\varphi_0), a_i(f_0,\theta_1,\varphi_0), \ldots, \\ a_i(f_{M-1},\theta_1,\varphi_0), \ldots, a_i(f_{M-1},\theta_{N-1},\varphi_0), \\ a_i(f_0,\theta_0,\varphi_1), \ldots, a_i(f_{M-1},\theta_{N-1},\varphi_{K-1}) \end{bmatrix} \tag{15}$$

$[]^\text{T}$ represents the transpose operation.

### 2.3. Observation Geometry Transformation for Space-Based Radar

In this section, the observation geometry transformation for a space-based radar is derived to obtain accurate observing angles, which are significant for GTD scattering center estimation and the HRRP prediction process.

As depicted in Figure 1, the Earth-centered, Earth-fixed (ECEF) coordinate system $O_e - X_eY_eZ_e$ and the radial tangential normal (RTN) coordinate system $O_t - X_tY_tZ_t$ are marked in black lines and in red lines, respectively. The origin of the ECEF coordinate system is at the center of the earth, and it is a non-rotating right-handed Cartesian coordinate system. The direction of its $X_e$ axis points to the equator, and the prime meridian

intersection and the $Z_e$ axis pass through the Earth's north pole. As for the RTN coordinate system, the $X_t$ axis is parallel to the direction of the velocity, the $Z_t$ axis is along the main body of the space target, and the $Y_t$ axis is determined according to the right-hand theorem.

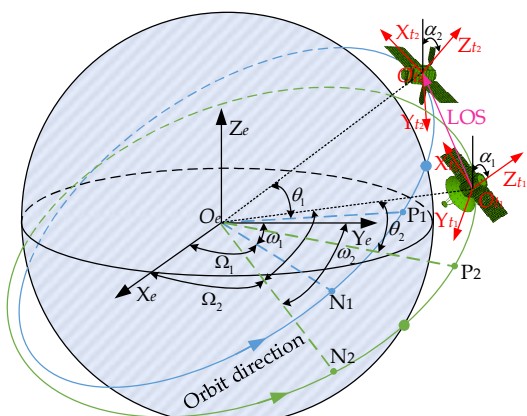

**Figure 1.** Schematic of the coordination system transformation.

The orbit of the space target is determined by six orbital elements, which are derived from a priori two-line elements (TLEs). To be specific, these elements are expressed as $h$ (semi-major axis), $\alpha$ (inclination), $\Omega$ (right ascension of the ascending node), $e$ (eccentricity), $\omega$ (argument of perigee), and $\theta$ (true anomaly). Thereafter, the observation geometry transformation between the ECEF coordinate system and the RTN coordinate system can be derived according to the given six orbital elements.

Then, the positions of the space target and the space-based radar in the ECEF coordinate system are given as

$$O_{t_1\,ECEF} = R_z(\Omega_1)R_x(\alpha_1)R_z(\omega_1)$$
$$\cdot \left[ h_1\frac{e_1+\cos\theta_1}{1+e_1\cos\theta_1} \quad \left(h_1\frac{1-e_1^2}{1+e_1\cos\theta_1}\right)\sin\theta_1 \quad 0 \right]^{\mathrm{T}} \tag{16}$$

$$O_{t_2\,ECEF} = R_z(\Omega_2)R_x(\alpha_2)R_z(\omega_2)$$
$$\cdot \left[ h_2\frac{e_2+\cos\theta_2}{1+e_2\cos\theta_2} \quad \left(h_2\frac{1-e_2^2}{1+e_2\cos\theta_2}\right)\sin\theta_2 \quad 0 \right]^{\mathrm{T}} \tag{17}$$

where $R_x$, $R_y$, and $R_z$ represent the rotation matrix around the $x$ axis, the $y$ axis, and the $z$ axis, respectively.

Also, the rotation matrices can be expressed as follows:

$$R_z(\Omega) = \begin{bmatrix} \cos\Omega & -\sin\Omega & 0 \\ \sin\Omega & \cos\Omega & 0 \\ 0 & 0 & 1 \end{bmatrix} \tag{18}$$

$$R_x(\alpha) = \begin{bmatrix} 1 & 0 & 0 \\ 0 & \cos\alpha & -\sin\alpha \\ 0 & \sin\alpha & \cos\alpha \end{bmatrix} \tag{19}$$

$$R_z(\omega) = \begin{bmatrix} \cos\omega & -\sin\omega & 0 \\ \sin\omega & \cos\omega & 0 \\ 0 & 0 & 1 \end{bmatrix} \tag{20}$$

Thus, the coordinate transformation matrix between the ECEF coordinate system and the RTN coordinate system can be expressed as

$$M_{ECEF-OF} = [R_z(\Omega)R_x(\alpha)R_z(\omega+\theta)]^{-1} \tag{21}$$

So, the positions of the space target and space-based radar in the RTN coordinate system are given as

$$O_{t_1 OF} = M_{ECEF-OF}(\Omega_1, \alpha_1, \omega_1, \theta_1)O_{t_1 ECEF} \tag{22}$$

$$O_{t_2 OF} = M_{ECEF-OF}(\Omega_1, \alpha_1, \omega_1, \theta_1)O_{t_2 ECEF} \tag{23}$$

Also, the line of sight (LOS) of the radar in the RTN reference coordinate system is computed as

$$LOS = O_{t_1 OF} - O_{t_2 OF} = (x_\Delta, y_\Delta, z_\Delta) \tag{24}$$

$$\theta_\Delta = \arctan\left(\frac{x_\Delta}{y_\Delta}\right) \tag{25}$$

$$\varphi_\Delta = \arctan\left(\frac{z_\Delta}{\sqrt{x_\Delta^2 + y_\Delta^2 + z_\Delta^2}}\right) \tag{26}$$

where $x_\Delta, y_\Delta, z_\Delta$ represent the location vector of the LOS in the RTN coordinate system and $\theta_\Delta$ and $\varphi_\Delta$ are the observing azimuth angle and elevation angle in the RTN coordinate system, respectively.

By deriving the observation geometry transformation for the space-based radar, the inherent motion of space targets can be eliminated from consideration in this paper.

## 3. Proposed Method

In Section 3.1, to predict the HRRP sequence more accurately, we first introduce a novel scattering parameter estimation method based on an improved 3D-ESPRIT algorithm. Then, in Section 3.2, HRRP characteristics obtained via the wide-band radar and HRRP pseudo-likelihood predicted by using the GTD scattering center model are employed in the PHD filter to fully utilize space targets' electromagnetic characteristics. Finally, the main steps of our proposed HGI-PHD filter are given in Section 3.3. To be specific, the flowchart of our proposed HGI-PHD filter is shown in Figure 2.

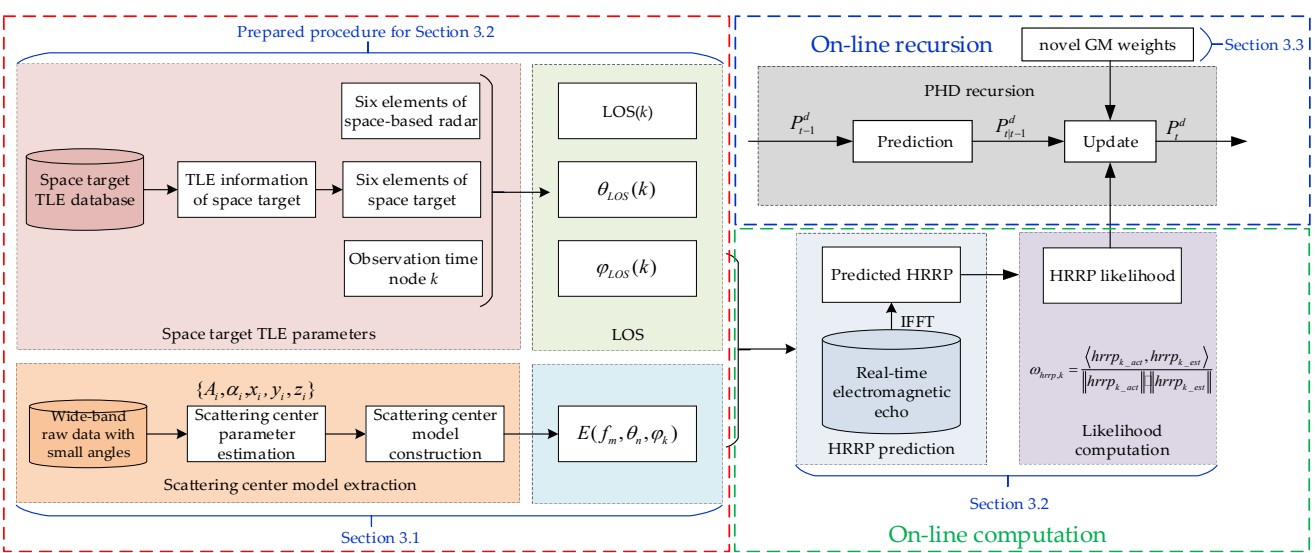

**Figure 2.** Flowchart of our proposed HGI-PHD filter.

### 3.1. Parameter Estimation of Scattering Centers

In this section, in order to predict the HRRP sequence more accurately, an improved 3D-ESPRIT algorithm with reconstructed covariance matrices is proposed to estimate

the 3D-GTD model parameters, and the main contribution of our proposed 3D-ESPRIT algorithm is shown as follows.

Firstly, by performing the spatial smoothing operation in the $x$ direction, a Hankel matrix pencil $X^x$ can be given by reference [27]

$$X^x = \begin{bmatrix} X_0^x & X_1^x & \cdots & X_{M-P}^x \\ X_1^x & X_2^x & \cdots & X_{M-P+1}^x \\ \vdots & \vdots & \cdots & \vdots \\ X_{P-1}^x & X_P^x & \cdots & X_{M-1}^x \end{bmatrix} \tag{27}$$

where

$$X_m^x = \begin{bmatrix} x(m,0) & x(m,1) & \cdots & x(m,K-L) \\ x(m,1) & x(m,2) & \cdots & x(m,K-L+1) \\ \vdots & \vdots & \cdots & \vdots \\ x(m,L-1) & x(m,L) & \cdots & x(m,K-1) \end{bmatrix} \tag{28}$$

$$x(m,k) = \begin{bmatrix} E(m,0,k) & E(m,1,k) & \cdots & E(m,N-Q,k) \\ E(m,1,k) & E(m,2,k) & \cdots & E(m,N-Q+1,k) \\ \vdots & \vdots & \cdots & \vdots \\ E(m,Q-1,k) & E(m,Q,k) & \cdots & E(m,N-1,k) \end{bmatrix} \tag{29}$$

where $P \in [I+1, M-I+1]$, $Q \in [I+1, N-I+1]$, $L \in [I+1, K-I+1]$, and $M$, $N$, $K$, and $I$ represent the frequency steps, azimuth angle steps, elevation angle steps, and the scattering center numbers, respectively.

Then, a permutation matrix $J$ is defined as

$$J = \begin{bmatrix} 0 & \cdots & 0 & 1 \\ \vdots & 0 & 1 & 0 \\ 0 & \cdot^{\cdot^{\cdot}} & \ddots & \vdots \\ 1 & 0 & \cdots & 0 \end{bmatrix}_{PQL \times PQL} \tag{30}$$

By combining the permutation matrix $J$ and the original back-scattering electromagnetic data $X^x$, a new matrix $E_{conj}$ containing the covariance information of the original scattered data $X^x$ is obtained by

$$E_{conj} = J \cdot X^x \tag{31}$$

Then, we construct the following three covariance matrices to fully utilize the electromagnetic scattering characteristic of radar targets:

$$\begin{cases} R_{X_x X_x} = X_x X_x^{\mathrm{H}} \\ R_{E_{conj} E_{conj}} = E_{conj} E_{conj}^{\mathrm{H}} \\ R_{X_x E_{conj}} = X_x E_{conj}^{\mathrm{H}} \end{cases} \tag{32}$$

Next, by averaging the above three covariance matrices, a novel covariance matrix $R$ is proposed in this paper, which is shown as follows:

$$R = \frac{R_{X_x X_x} + R_{E_{conj} E_{conj}} + R_{X_x E_{conj}}}{3} \tag{33}$$

Note that $R$ is a Hermitten matrix, and it satisfies $R = R^{\mathrm{H}}$. Thus, by squaring matrix $R$, we have the final covariance matrix $R_1$ as follows:

$$R_1 = RR^{\mathrm{H}} = R^2 \tag{34}$$

Also, the following relation satisfies

$$\begin{cases} \lambda_{1i} = \lambda_i{}^2 \\ \mathbf{\Lambda}_{1i} = \mathbf{\Lambda}_i \end{cases} \quad i = 1, \dots, I \tag{35}$$

where $\lambda_{1i}$ and $\mathbf{\Lambda}_{1i}$ are the eigenvalue and the eigenvector of $\boldsymbol{R}_1$, respectively, and $\lambda_i$ and $\mathbf{\Lambda}_i$ are the eigenvalue and the eigenvector of $\boldsymbol{R}$, respectively.

Note that from (34), the eigenvalues of $\boldsymbol{R}_1$ are twice those of $\boldsymbol{R}$. Therefore, the differences between the noise eigenvalues and the signal eigenvalues can be broadened by constructing $\boldsymbol{R}_1$.

To analyze our motivation for constructing the aforementioned covariance matrix mathematically, the estimated parameters variance is derived as follows:

$$E\left\{(\hat{\xi} - \xi)^2\right\} = \frac{\frac{\sigma^2}{2MNK} \sum\limits_{i=1}^{I} -\frac{\gamma_i}{(\sigma^2 - \gamma_i)^2} \left| \boldsymbol{G}^{\mathrm{H}} \cdot (v_i)^{\mathrm{H}} \right|^2}{\sum\limits_{i=I+1}^{MNK} -\left| \left[ d\boldsymbol{G}^{\mathrm{H}}/d(z) \right] \cdot (v_i)^{\mathrm{H}} \right|^2} \tag{36}$$

where $E\left\{(\hat{\xi} - \xi)^2\right\}$ represents variance of the estimated parameters; $\hat{\xi}$ and $\xi$ represent the estimated parameter and the original parameter, respectively; $\sigma^2$ and $\gamma_i$ denote eigenvalues of noises and eigenvalues of signals, respectively; $v_i = \gamma_i \boldsymbol{I} - \boldsymbol{X}^x$ represents the eigenmatrix of $\gamma_i$; and $\boldsymbol{I}$ denotes a identify matrix.

It can be seen from (36) that $E\left\{(\hat{\xi} - \xi)^2\right\}$ decreases when $\sigma^2$ and $\gamma_m$ differ greatly from each other, which will bring out more accurate scattering center parameters. So, constructing $\boldsymbol{R}$ can broaden the differences between $\sigma^2$ and $\gamma_m$, which can estimate the GTD model parameters more accurately.

Then, the scattering parameters $\{\alpha_i, x_i, y_i, z_i\}$ are estimated according to reference [27], and the intensity parameters $\tilde{A}$ can be obtained by using the least square method in (27). It should be noticed that for the sake of readability, the detailed CRB derivation of the GTD scattering center parameter is given in Appendix A.

*3.2. HRRP Characteristic-Aided Filter*

Once we have estimated the scattering center model parameters, the HRRP sequence can be predicted at a relatively small observing angle. Thus, the predicted HRRP information can be utilized in the update recursion of the PHD filter. In this section, we will introduce the specific implementation of our proposed HRRP characteristic-aided filter.

Let us denote the surveillance region as three-dimensional; then, the target state can be expressed as

$$\boldsymbol{x}_k = \left[ l_{x,k}, v_{x,k}, l_{y,k}, v_{y,k}, l_{z,k}, v_{z,k} \right]^{\mathrm{T}} \tag{37}$$

where at time $k$, $l_{x,k}$, $l_{y,k}$, and $l_{z,k}$ are the three-dimensional locations of targets. $v_{x,k}$, $v_{y,k}$, and $v_{z,k}$ are the three-dimensional velocities of targets.

Thereafter, as the HRRP characteristics are employed, the measurement state of targets at time $k$ yields

$$\boldsymbol{z}_k = \left[ z_{l,k}, z_{hrrp,k} \right] \tag{38}$$

where at time $k$, $z_{l,k}$ and $z_{hrrp,k}$ denote the location measurements and the HRRP measurements, respectively.

The specific prediction and update recursion of the proposed HGI-PHD filter are discussed as follows.

***Prediction:***

The prediction recursion of the proposed filter is written as

$$D_{k|k-1}(x_k) = \gamma_k(x_k) + \int \left( p_{s,k}(x_{k-1})f_{k|k-1}(x_k|x_{k-1}) + b_{k|k-1}(x_k|x_{k-1})D_{k-1|k-1}(x_{k-1}) \right)dx_{k-1}$$

(39)

where at time $k$, $\gamma_k(x_k)$ denotes the intensity of the birth RFS, $p_{s,k}(x_{k-1})$ is the survival probability, $f_{k|k-1}(x_k|x_{k-1})$ is the Markov transmission density, $b_{k|k-1}(x_k|x_{k-1})$ represents the density of the spawned at time $k$ under the condition of the previous state at time $k-1$, and $D_{k-1|k-1}(x_{k-1})$ is the density of the PHD filter at time $k-1$.

***Update*:**

After employing the HRRP characteristics in the standard PHD filter, the update recursion of the proposed HGI-PHD filter can be given by

$$D_{k|k}(x_k) = L_{\mathbf{Z}}(x_k) \cdot D_{k|k-1}(x_k)$$

(40)

Assuming the HRRP characteristic of targets is independent of location states, the likelihood for targets and for clutter can be denoted as

$$g(z|x) = g((z_{l,k}, z_{hrrp,k})|x) = g_l(z_l|x)g_{hrrp}(hrrp)$$

(41)

$$c(z|x)) = c_z(z|x)c_{hrrp}(hrrp)$$

(42)

where $g_l(z_l|x)$ and $g_{hrrp}(hrrp)$ are the state location likelihood and HRRP pseudo-likelihood functions for targets, respectively. $c_z(z|x)$ and $c_{hrrp}(hrrp)$ are the state location likelihood and HRRP pseudo-likelihood functions for clutter, respectively.

However, how do we predict the HRRP sequence as time varies, and how do we compute the HRRP pseudo-likelihood for various targets and clutter? These two problems are core points to be solved in this paper, and we give the detailed process as follows.

(1)    *HRRP sequence prediction*

Note that the radar is able to provide range and bearing measurements of the target's center, and the observation model of kinematic measurement is denoted as follows:

$$M_k = [R_k, \beta_k] = h(l_k) + w_k$$

(43)

where at time $k$, $h(\cdot)$ denotes the kinematic observation function; $R_k$ and $\beta_k$ represent the target range measurements and bearing measurements, respectively; and $w_k$ represents the zero-mean observation noise matrix.

For the space target, due to its previously known orbital information and low maneuverability, the velocity direction is almost aligned with the axial direction of the target's main body, which is shown in Figure 3.

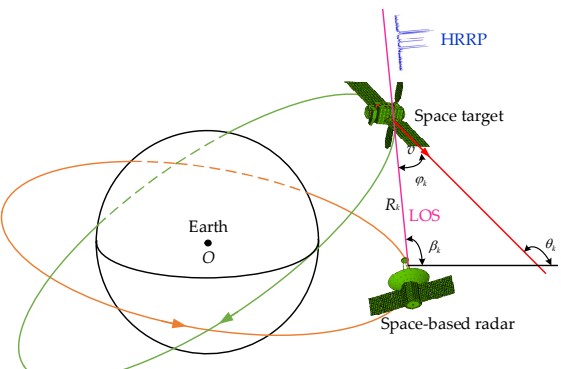

**Figure 3.** Schematic of target observing angle.

Therefore, the observing angle can be computed as

$$\varphi_k = \theta_k - \beta_k = \cos^{-1}\left(\frac{x_k v_{x,k} + y_k v_{y,k}}{\sqrt{x_k^2 + y_k^2}\sqrt{v_{x,k}^2 + v_{y,k}^2}}\right) \tag{44}$$

where $\theta_k = \tan^{-1}\left(\frac{v_{y,k}}{v_{x,k}}\right)$ denotes the heading angle for space targets.

Once the specific observing angles are given as time varies, different back-scattering echoes can be obtained via the 3D scattering center model in (6). Then, the frequency response of the $m$-th frequency point can be written as $E_m = E(f_m, \theta_n, \varphi_k)$, and by taking an inverse discrete Fourier transform (IDFT) on frequency response sequence $E = E_0, E_1, \ldots E_M$, the HRRP sequence of space targets can be predicted eventually, which is denoted as $hrrp_{k\_est}$ in this paper.

(2)  *HRRP pseudo-likelihood evaluation*

At time $k$, we can acquire the actual HRRP sequence of different space targets or clutter, which is denoted as $hrrp_{k\_act}$. Then, the HRRP pseudo-likelihood can be computed as

$$g_{hrrp,k} = \frac{\langle hrrp_{k\_act}, hrrp_{k\_est}\rangle}{\|hrrp_{k\_act}\|\cdot\|hrrp_{k\_est}\|} \tag{45}$$

where $\langle a, b\rangle = \int a(x)b(x)dx$ represents the inner product operation and $\|h\|$ is the $l_2$ norm function.

Note that the posterior intensity at time $k-1$ is in the form of a Gaussian mixture; then, we have

$$v_{k|k-1}(x) = \sum_{i=1}^{J_{k|k-1}} w_{k|k-1}^i \mathcal{N}\left(x; m_{k-1}^i, P_{k-1}^i\right) \tag{46}$$

where $w_{k|k-1}^i$ is the $i$-th Gaussian mixture weight at time $k-1$; $m_{k-1}^i$ and $P_{k-1}^i$ denote the mean vector and covariance vector of the $i$-th Gaussian mixture weight at time $k-1$, respectively; and $J_{k|k-1}$ represents the number of Gaussian mixture weights at time $k-1$.

Substituting (46) into (40), we have the posterior intensity at time $k$ as

$$D_{k|k}(x_k) = [1 - p_d]D_{k|k-1}(x_k) + \sum_{i=1}^{J_{k|k-1}} \sum_{z_k \in Z} \widetilde{w}_{k|k-1}^i \mathcal{N}\left(x_k; m_{k|k-1}^i, P_{k|k-1}^i\right) \tag{47}$$

where the normalized GM weight is denoted as

$$\widetilde{w}_{k|k-1}^j(z_k) = \frac{p_{D,k}(x)g_z(z|x)g_{hrrp}(hrrp)w_{k|k-1}^j q_k^j(z_k)}{\kappa(z)c_{hrrp}(hrrp) + p_{D,k}(x)g_z(z|x)g_{hrrp}(hrrp)\sum\limits_{i=1}^{J_{k|k-1}} w_{k|k-1}^i q_k^i(z_k)} \tag{48}$$

where $\begin{cases} q_k^i(z_k) = N\left(z_k, \eta_{k|k-1}^i, S_{k|k-1}^i\right) \\ \eta_{k|k-1}^i = H_k m_{k|k-1}^i \\ S_{k|k-1}^i = H_k P_{k|k-1}^i H_k^T + R_k \end{cases}$ .

*3.3. Improved Gaussian Component Weight Merging Principal*

By utilizing (43), the posterior intensity of targets can be obtained, and different intensities are represented by different constituent Gaussian component weights. With the aim of tracking space targets with high clutters effectively, label $\kappa$ is denoted in this paper. Then, we have the set $v_k = \left\{w_k, x_k^i, P_k^i, \kappa_k^i\right\}$, and $\kappa_k^i$ denotes the label of the $k$th target at time $i$.

Following the RFS scheme, the constituent Gaussian mixture component weights are modified as follows.

(1) First, the index of the highest weight component in the target posterior density is selected as follows:

$$i_{\max} = \underset{i \in I}{\arg\max}(w_k^i) \tag{49}$$

where $I$ is the index set.

(2) Then, the set of component orders that are most similar to the largest weighted component in the target posterior intensity at time k is represented as

$$\Phi_k = \left\{ i \mid \left( \left( x_k^i - x_k^{i_{\max}} \right)^{\mathrm{T}} \left( P_k^i \right)^{-1} \left( x_k^i - x_k^{i_{\max}} \right) \leq U_{mm} \right) \right\} \tag{50}$$

$$U_{mm} = \left( 1 + w_k^i \right) \sigma_s \tag{51}$$

where $\sigma_s$ is the variance of the measured Gaussian white noise.

(3) If the Gaussian component satisfies Equation (50) and the sum of the component weights fails to exceed twice the state extraction threshold, then a novel set $v_k^\tau$, which contains components, can be obtained as follows:

$$\widetilde{l}_k^\tau = i_k^{i_{\max}}, \widetilde{w}_k^\tau = \sum_{i \in \Phi_k} w_k^i, x_k^\tau = \frac{1}{\widetilde{w}_k^\tau} \sum_{i \in \Phi_k} w_k^i x_k^i \tag{52}$$

While $\sum\limits_{k=1}^{M_k} \widetilde{w}_k^\tau > 1$, the sub-weights of the target are multiplied by a penalized factor except the highest weight $i_{\max}$. Therefore, the novel weights yield the following:

$$w_{novel}^k = \begin{cases} \widetilde{w}_k^\tau, if\, k = i_{\max} \\ \beta \widetilde{w}_k^\tau, if\, k \neq i_{\max}, k \in [1, \ldots, M_k] \end{cases} \tag{53}$$

where $\beta = \alpha(1 - w_k^{i_{\max}})$ denotes the penalized factor and $0 \leq \alpha \leq 1$ is a constant that can determine the penalized degree.

(4) Substituting (53) into (48) yields

$$\overline{w}_{novel}^{i,n} = \begin{cases} \dfrac{w_k^{i,n}}{\kappa_k(\lambda_c) + w_k^{i_{\max}} \beta + \sum\limits_{i=1}^{N_{k|k-1}} w_k^{i,n}}, i \neq i_{\max}, i \in [1, M_k], n \in [1, N_{k|k-1}] \\[4mm] \dfrac{w_k^{i_{\max}, n} \beta}{\kappa_k(\lambda_c) + w_k^{i_{\max}} \beta + \sum\limits_{i=1}^{N_{k|k-1}} w_k^{i,n}}, i = i_{\max}, n \in [1, N_{k|k-1}] \end{cases} \tag{54}$$

where $w_k^{n,m} = p_{D,k}(x) g_{rcs}(rcs) g_d(d) w_{k|k-1}^j q_k^j(z_k)$ and $\kappa_k(\lambda_c) = \lambda_c c_{rcs}(rcs) c_d(d)$.

(5) Finally, the novel posterior density is computed as follows:

$$v(x) = \sum_{J_k}^{\tau=1} w_k^\tau N\left( x; x_k^\tau, p_k^\tau \right) \tag{55}$$

$$\widetilde{P}_k^\tau = \frac{1}{x_k^\tau} \sum_{i \in \Phi_k} w_k^i \left( P_k^i + \left( x_k^\tau - x_k^i \right) \left( x_k^\tau - x_k^i \right)^{\mathrm{T}} \right) \tag{56}$$

We can observe from (51) that $U_{mm}$ is an adaptive threshold varying with the measured noise in tracking scenarios. Therefore, the threshold can be dynamically adjusted during the whole tracking process and has a better tracking performance for the closely spaced targets.

### 3.4. Key Steps of the Proposed HGI-PHD Filter

The main steps of our proposed methods are given as follows. For the sake of readability, the corresponding pseudo-code of our proposed HGI-PHD filter is provided in Appendix B.

***Step 1*** Calculating the time-varying line-of-sight (LOS) and observing angles.

***Step 2*** Estimating the GTD scattering parameter via our proposed improved 3D-ESPRIT algorithm and reconstructing the scattering center model.

***Step 3*** Combining the time-varying observing angles with the scattering center model to predict the real-time scattering echo of space targets and applying an IFFT transform to the real-time electromagnetic echo to obtain the predicted HRRP sequence.

***Step 4*** Computing the HRRP likelihood for different targets and clutter.

***Step 5*** Improving the constituent Gaussian component weights by using the method introduced in Section 3.3.

***Step 6*** Employing the HRRP likelihood and the improved Gaussian component weight merging principal (e.g., $\overline{w}_{novel}^{i,n}$) in the update recursion of the PHD filter.

## 4. Simulation Results

In this paper, a merging PHD (HGI-PHD) filter aided by HRRP characteristics and improved with Gaussian mixture components is proposed, which can be applied to track space targets accurately. The time-varying HRRP pseudo-likelihood is utilized in the GM process of the standard PHD filter. The performances of our proposed HGI-PHD filter are validated by the following simulation experiments. Furthermore, the capability of the proposed HGI-PHD filter to discriminate approaching space targets is also demonstrated by utilizing the star-link satellites' real raw data.

### 4.1. Performance Evaluation of Scattering Center Estimation

In this section, the performance of the modified algorithm with respect to the SNR is compared with the other two methods. We set the scattering center parameters as shown in Table 1; the matrix beam parameters $P$, $Q$, and $L$ are all set as 6; the SNR varies from 0 dB to 30 dB with an interval of 1 dB; and 200 Monte Carlo trials are performed for each fixed *SNR*. To simplify, here, we just provide the mean RMSE of the scattering centers obtained by using different methods. Simulation results are depicted in Figure 4. It can be observed from Figure 4a–e that all of the scattering parameter estimation accuracies obtained via different methods increase as SNR increases, and our proposed improved 3D-ESPRIT method has a better parameter estimation performance than the traditional 3D-ESPRIT algorithm, the method in reference [15], and the method in reference [27]. Furthermore, note that the location parameter estimation accuracy is better than that of the type parameter and intensity parameter due to the latter two parameters being dependent on the aforementioned location parameters.

**Table 1.** Scattering parameters.

| Scattering Centers | $x_i(m)$ | $y_i(m)$ | $z_i(m)$ | $\alpha_i$ | $A_i$ |
|---|---|---|---|---|---|
| Scattering center 1 | 1.202 | 1.080 | 1.320 | 1.0 | 6.2 |
| Scattering center 2 | 1.353 | 1.263 | 1.541 | 0.5 | 5.6 |
| Scattering center 3 | 1.534 | 1.702 | 2.520 | 0 | 4.7 |
| Scattering center 4 | 1.892 | 2.322 | 3.210 | 1.0 | 3.4 |

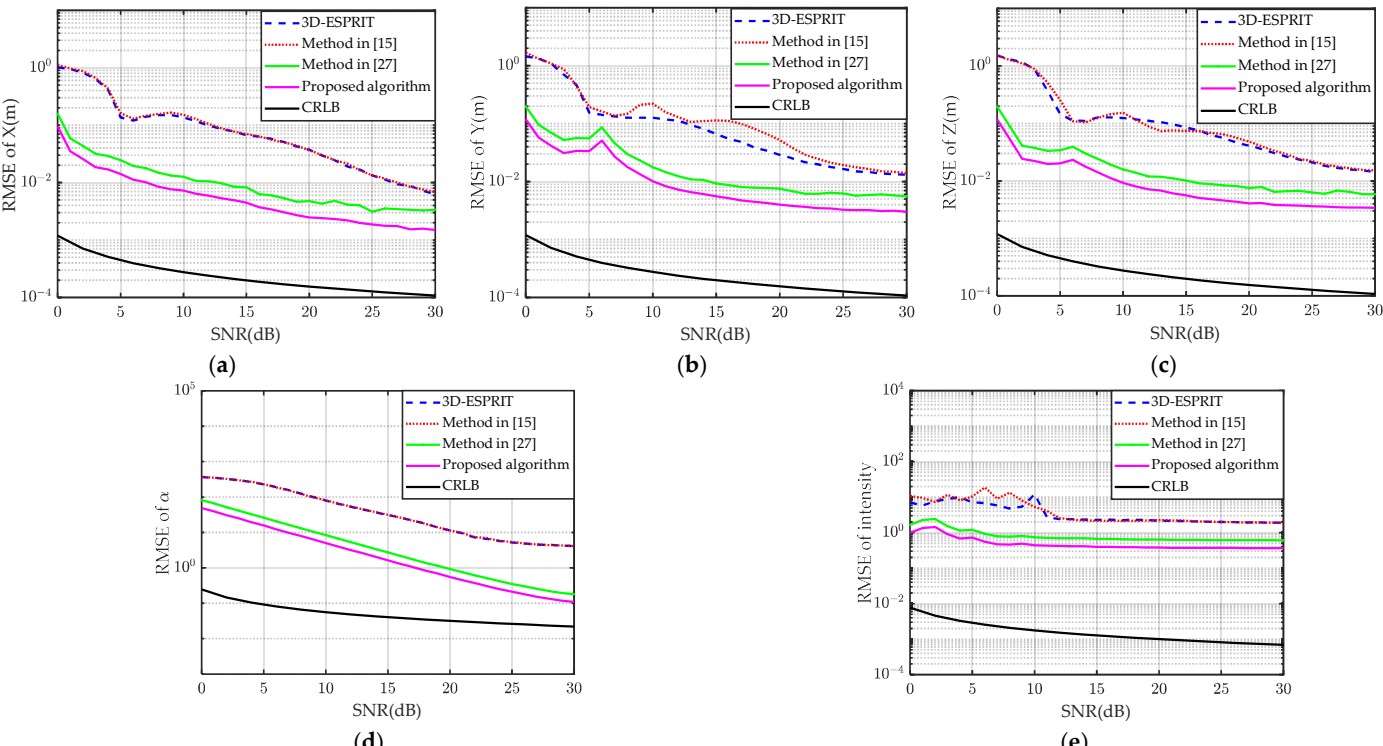

**Figure 4.** Comparisons of scattering parameter estimation accuracy between different methods. (**a**) X, (**b**) Y, (**c**) Z, (**d**) Type, and (**e**) Intensity.

### 4.2. HRRP Sequence Reconstructed by the GTD Model

To evaluate the reconstruction and parameter estimation performance of the proposed improved 3D-ESPRIT algorithm with simulated data, the back-scattering electromagnetic echo of three typical targets is calculated using electromagnetic computing software. To be specific, the shape and geometrical size of these three typical targets are depicted in Figure 5. In this simulation, we set the frequency number as 101 points with a range of 15.7~17.7 GHz and a frequency step of 20 MHz. Here, the incident azimuth angle and the elevation angle are set as 0° and 90°, respectively.

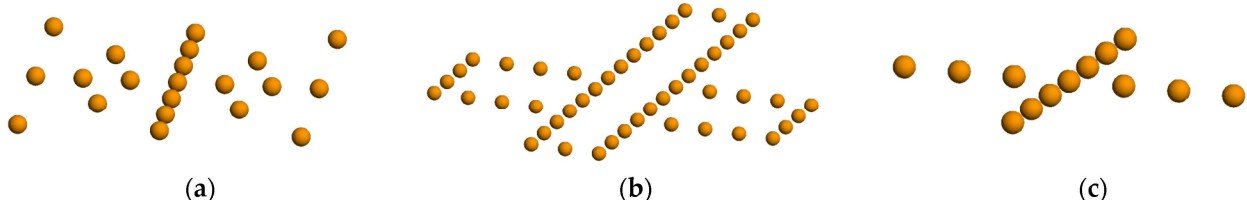

**Figure 5.** Scattering center model. (**a**) Target 1, (**b**) Target 2, and (**c**) Target 3.

It can be observed from Figure 6 that for these three typical targets, there are 3, 3, and 2 sharp peaks in the HRRP curve of the original data. Note that the reconstructed HRRP curve of these three targets obtained by using our improved 3D-ESPRIT algorithm matches the whole trend well, and it can also describe the fluctuation characteristics accurately. The above simulation results validate the effectiveness of our improved 3D-ESPRIT method and the feasibility of HRRP reconstruction via the GTD scattering center model.

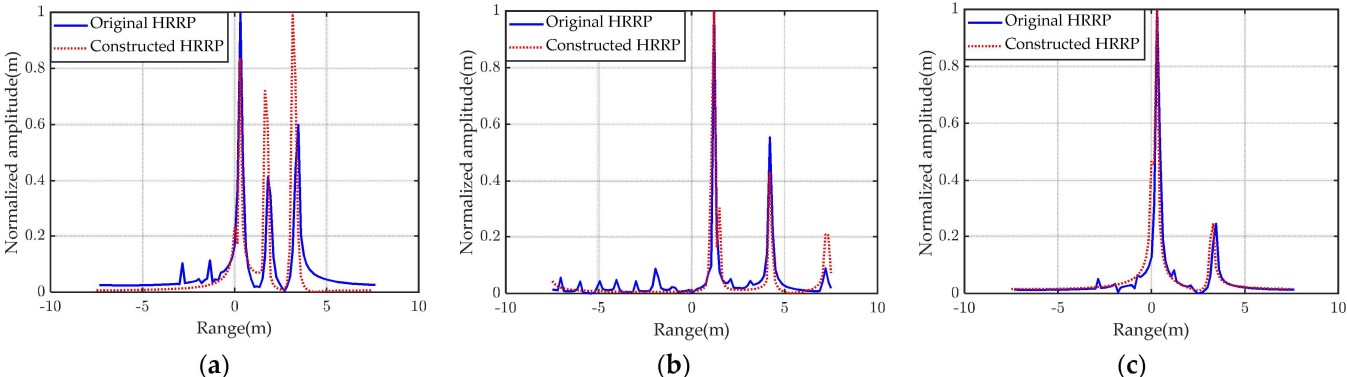

**Figure 6.** Reconstructed HRRP of three typical space targets with the GTD scattering center model when the azimuth angle is 0 and the elevation angle is 90. (**a**) Target 1, (**b**) Target 2, and (**c**) Target 3.

### 4.3. HRRP Similarity Rate Comparison

Firstly, the following three-dimensional (3D) simulation scenario is set to obtain the dynamic HRRP of three typical space targets, as depicted in Figure 7. From Figure 7, it can be seen that there are three typical space targets in the surveillance tracking region of the space-based radar.

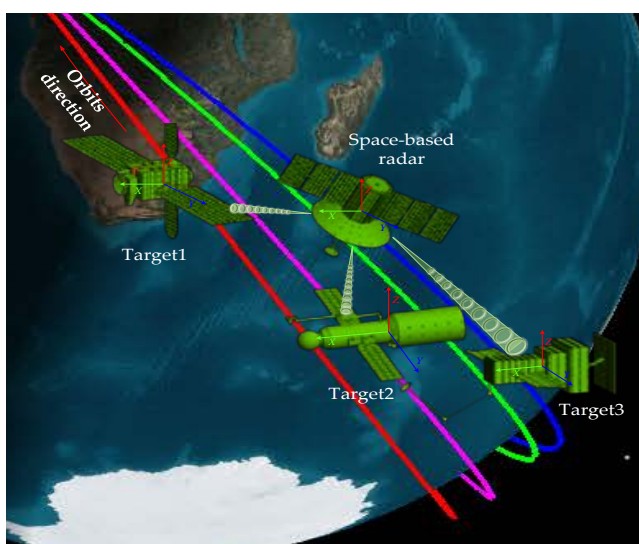

**Figure 7.** Three-dimensional simulation scenario.

In most applications, LOS parameter sequences are acquired via different methods. In this paper, we solve satellite targets with TLE ephemeris to compute their positions in the SGP4 model, and then their positions relative to the space-based radar can be calculated eventually. Hence, the LOS parameters can be calculated by using the spacecraft orbit computing software, which is shown in Figure 8. Table 2 gives the specific simulation parameters of the space radar and the three typical satellite targets in the spacecraft orbit computing software. Finally, combining the predicted HRRP sequence and the theoretical HRRP sequence of different targets and clutter, we obtain the dynamic coefficients among different targets and clutter as shown in Figure 9. It can be obviously seen that the predicted HRRP and the theoretical HRRP of the same target have the highest similarity rate than those of different targets or clutter. That means the same target has the largest HRRP pseudo-likelihood with its theoretical HRRP sequence, which can be utilized to improve the update accuracy and space target tracking performance.

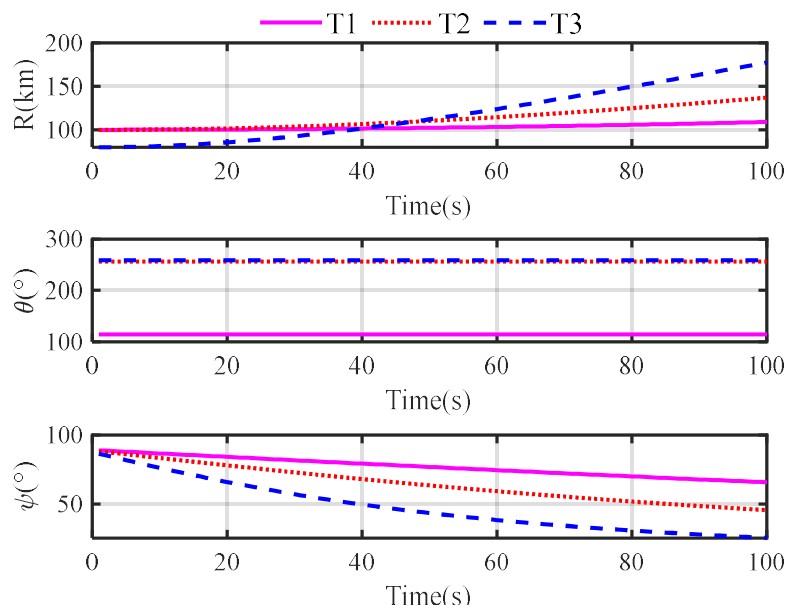

**Figure 8.** LOS parameters of three typical space targets. (T1: Target 1, T2: Target 2, T3: Target3).

**Table 2.** Simulation parameters in the spacecraft orbit computing software.

| Parameters | Value |
|---|---|
| Range of detection $R$ | 200 km |
| Orbit altitude of space-based radar $H_0$ | 500 km |
| Orbit altitude of target 1 $H_1$ | 400 km |
| Orbit altitude of target 2 $H_2$ | 500 km |
| Orbit altitude of target 3 $H_3$ | 480 km |
| Orbit inclination of space-based radar $\vartheta_0$ | 45° |
| Orbit inclination of target 1 $\vartheta_1$ | 48° |
| Orbit inclination of target 2 $\vartheta_2$ | 55° |
| Orbit inclination of target 2 $\vartheta_3$ | 60° |

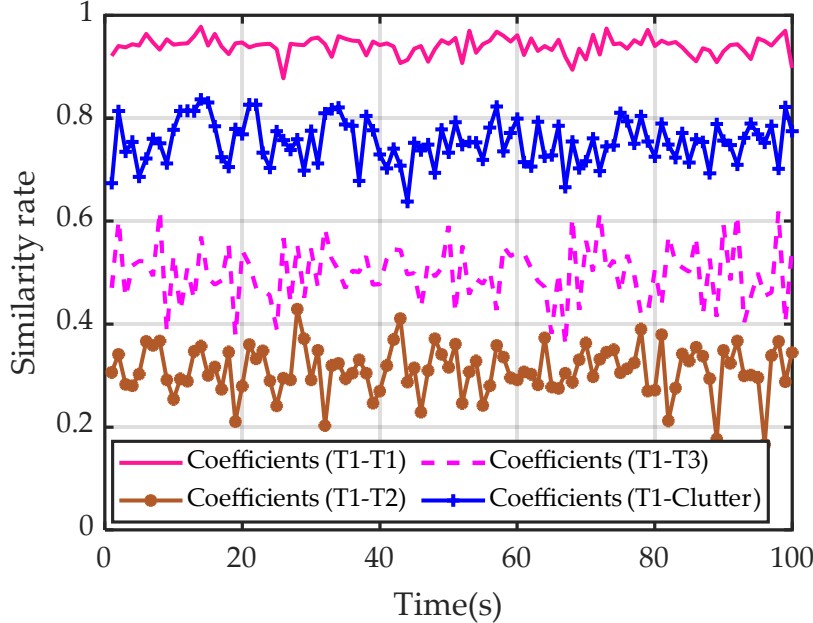

**Figure 9.** Dynamic coefficients among different targets and clutter.

*4.4. Comparisons of OSPA Errors*

Suppose there are three space targets existing in the surveillance region with a detection range of $[0, 100]$km $\times$ $[0, 100]$km $\times$ $[0, 100]$km. As for these three space targets, all of them obey the variable acceleration motion model, and the kinematic state vector includes the positions, velocities, and accelerations as $z_k = \left[ x, y, z, v_x, v_y, v_z, a_x, a_y, a_z \right]^{\text{T}}$. Moreover, it should be noticed that the measurement vector $\tilde{x}_k = [x_k, HRRP_k]^{\text{T}}$ includes the position measurements and HRRP characteristic measurements.

Let us set the duration for the whole surveillance simulation as 100 s and the survival probability for independent space targets as 0.98. As for the three space targets, they are born at 1s, and the second target vanishes at 70s, while the third one is present throughout the whole tracking process. Then, the pruning procedure is employed with varying time by utilizing the Gaussian weight threshold as $T_{th} = 10^{-5}$ and the maximum number of the mixture components as 100, and we fix the cardinality distribution over 100 terms. Furthermore, the detection probability is set as $P_d = 0.98$ for all filters, $V = 8 \times 10^6$ m$^3$ represents the whole space surveillance region volume, and measurements for detected targets are immersed with clutter, which obeys a Poisson RFS distribution and has an intensity of $1.4 \times 10^{-5}$ m$^{-3}$ (i.e., 112 false alarms over space surveillance region per frame). Notice that the HRRP probability density functions of clutter are obtained by adding white Gaussian noise to the HRRP probability density functions of targets.

Here, the optimal sub-pattern assignment (OSPA) metric is employed for the detection performance evaluation metric, and its specific form is given as follows [28]:

$$D_c^p(X, Y) \triangleq \left[ \frac{1}{N} \left( \min_{\ell \in \Pi_N^M} \sum_{i=1}^M d_c(x_i, y_{\ell(i)})^p + c^p(N - M) \right) \right]^{\frac{1}{p}} \qquad (57)$$

where $d_c(x_i, y_{\ell(i)}) \triangleq \min(c, \|x_i - y_{\ell(i)}\|)$ is the cut-off distance with parameter $c > 0$, $\|\cdot\|$ denotes the Euclidean distance, and $\Pi_N^M$ represents the set of permutations of cardinality $M$ on $\{1, 2, \ldots, N\}$. $p$ is the order parameter of the OSPA metric with a range of $1 \le p \le \infty$, and the OSPA distance is given by $d_c^p(X, Y) = d_c^p(Y, X)$, which corresponds to the case of $M > N$. Then, the average OSPA results after 100 Monte Carlo simulations for detection probabilities with parameters set as $p = 1$ and $c = 200$ m are shown in Figure 10, which is able to demonstrate the performance metrics for the accuracy of target location estimation as well as target number estimation.

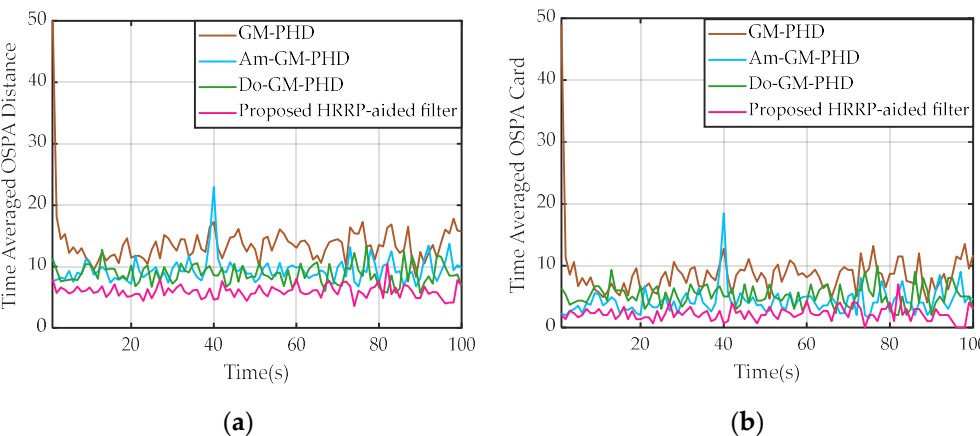

(**a**)                    (**b**)

**Figure 10.** Average of OSPA errors over 100 Monte Carlo trials. (**a**) Mean OSPA distance. (**b**) Mean OSPA cardinality.

It can be seen from Figure 10a,b that the GM-PHD filter has the highest location error and average OSPA cardinality among the four methods in general. Note that for most of the observing time, the amplitude-aided GM-PHD (Am-GM-PHD) filter and the Doppler-aided

GM-PHD (Do-GM-PHD) filter have better location estimation accuracy and cardinality estimation accuracy than the conventional GM-PHD filter. We would like to mention that the estimated OSPA location obtained by the Am-GM-PHD filter fluctuates sharply as the time sequence varies. Furthermore, it can be observed that our proposed HRRP-aided filter has lower estimation OSPA results than the other three methods as depicted in Figure 11. We can also notice that our proposed filter fits better with the ground truth data than the other three methods.

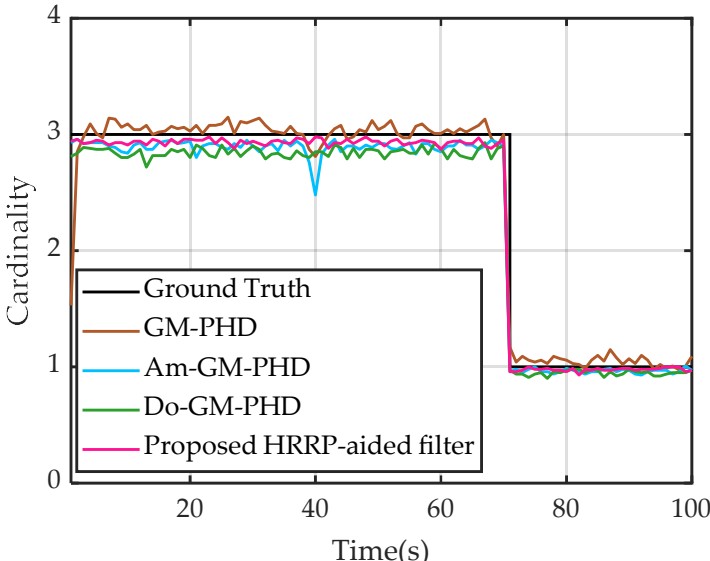

**Figure 11.** Cardinality estimation results.

The explanation is that, as expected, the proposed filter utilizes the HRRP characteristics of space targets obtained via the wide-band space-based radar system, which is more delicate in describing the target scattering features than the aforementioned ones (e.g., amplitude, etc.). Simulation results show that the proposed HRRP-aided filter outperforms the GM-PHD filter, the Am-GM-PHD filter, and the Do-GM-PHD filter, especially in dense clutter detection scenarios.

*4.5. Effective Tracking of Two Closely Spaced Targets*

To further exploit the performance of our proposed HGI-PHD filter to track closely spaced multiple targets, two closely spaced targets are chosen for tracking in this subsection. Note that due to Space-X satellites having similar orbit characteristics, two star-link satellites with serial numbers 55713U and 55715U are selected as the closely spaced targets to validate the discrimination functionality of the proposed filter.

The TLE information and observing duration for the two selected star-link satellites are depicted in Table 3, and the observing duration is from 3 Mar 2023 01:55:39.000 to 3 Mar 2023 01:57:19.000. Moreover, the 3D observation diagram in the spacecraft orbit computing software is shown in Figure 12.

**Table 3.** TLE information of the two selected star-link satellites.

| Targets | TLE Information |
|---|---|
| Satellite 1 | 1 55713U 23026U 23066.91667824 .00087040 00000-0 95406-3 0 9990<br>2 55713 43.0022 235.7401 0001445 276.7526 56.7822 15.62262486 1301 |
| Satellite 2 | 1 55715U 23026W 23066.91667824 .00053204 00000-0 58720-3 0 9995<br>2 55715 41.0058 235.6894 0002065 272.0442 84.7138 15.62266923 1303 |

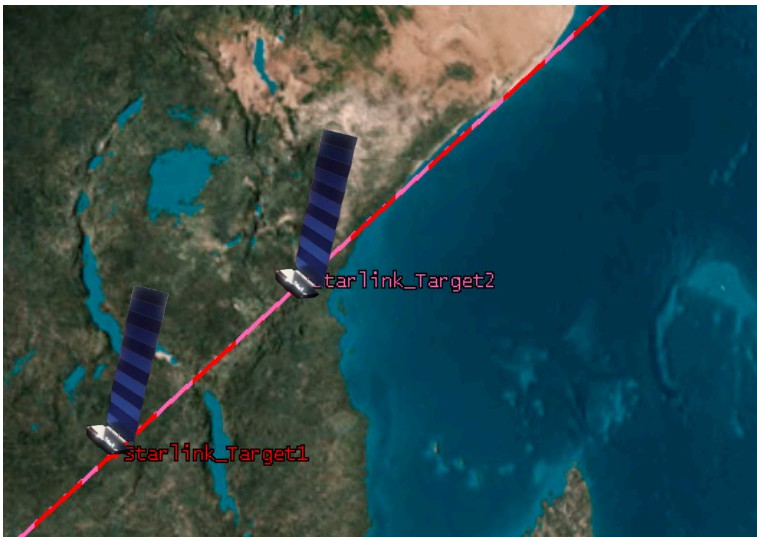

**Figure 12.** Three-dimensional observation diagram of two closely spaced targets.

After obtaining the trajectories of the two space targets, the tracking performance of various MTT filters can be compared. Figure 13a,b show the time-averaged OSPA distance and the time-averaged OSPA cardinality over 200 Monte Carlo trials, respectively. Figure 14 shows the mean deviations of the cardinality distributions. As shown in Figures 13 and 14, the proposed HGI-PHD filter produces the best results in the simulation not only in OSPA distance but in cardinality estimation as well. In contrast, the worst performance is from the standard GM-PHD filter, i.e., both its OSPA distance and OSPA cardinality are high in the initial stage of our simulation. Furthermore, it can be also seen that the results obtained from the Am-GM-PHD filter and the Do-GM-PHD filter fluctuate more violently than those obtained from the HGI-PHD filter. Hence, we are convinced by comparing Figures 13 and 14 that the proposed HGI-PHD filter outperforms the other three filters for resolving closely spaced targets.

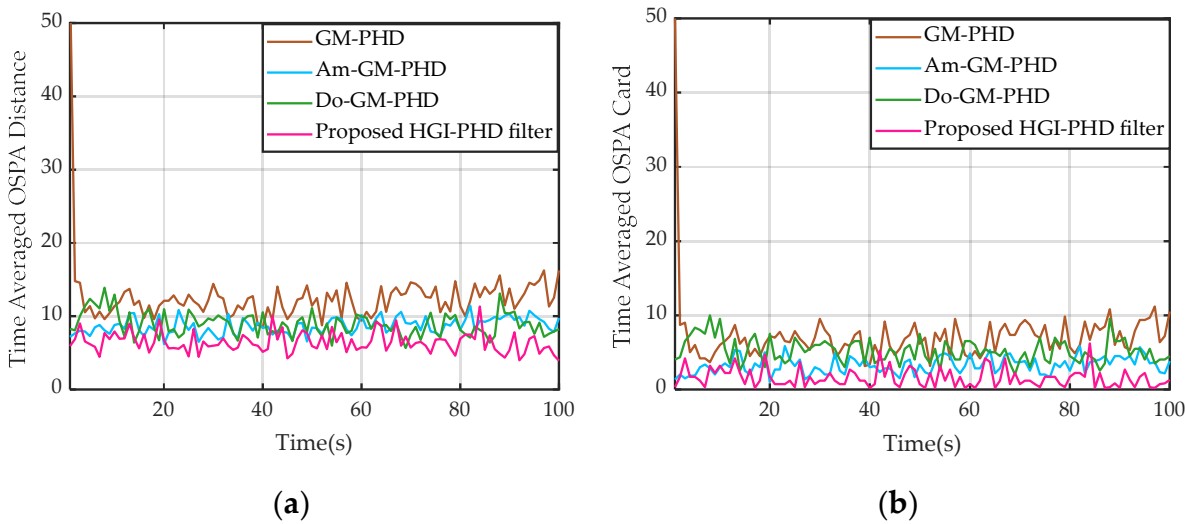

**Figure 13.** Tracking performance for two closely spaced targets. (**a**) Mean OSPA distance. (**b**) Mean OSPA cardinality.

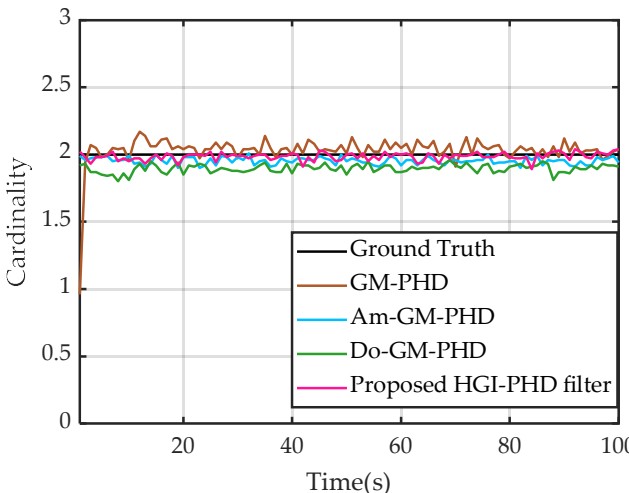

**Figure 14.** Cardinality estimation results of two closely spaced targets.

*4.6. Computational Time Analysis*

To analyze the computational complexity of different filters, the computational time is compared with an Intel(R) Core(TM) i7-8700 CPU @ 3.20GHz PC with 16 GB RAM and MATLAB R2021b. The core number we have utilized is 6, and the operating system is Windows 10. And the average computational time of the four filters is given in Table 4 after 100 Monte Carlo trials. It can be observed the proposed filter has a little heavier computational burden than the Am-GM-PHD filter under two clutter scenarios, which are 3.005 s and 3.086 s corresponding to two different clutter rates. The Do-GM-PHD filter and the standard GM-PHD filter have similar computational costs. In fact, due to computing the prior knowledge of the time-varying HRRP pseudo-likelihood, the proposed filter has a similar computational cost to the Am-GM-PHD filter theoretically, and the results in Table 4 validate it quantitatively.

**Table 4.** Comparisons of Average Computational Time.

| Clutter $\lambda_c$ | GM-PHD | Am-GM-PHD | Do-GM-PHD | The Proposed Filter |
|---|---|---|---|---|
| 50 | 0.725 s | 3.249 s | 0.780 s | 3.005 s |
| 100 | 0.815 s | 3.425 s | 1.075 s | 3.086 s |

**5. Conclusions**

In the standard PHD filter, the tracking performance deteriorates sharply when clutter rates are high, detection probabilities are low, or targets approach each other. To deal with the aforementioned problems, an adapted PHD-based filter aided by HRRP characteristics and with an adapting threshold for the GM component merging procedure was proposed in this paper. By fully using the HRRP information, the electromagnetic characteristics of space targets were utilized in the update recursion of the standard PHD filter. We have also derived the mathematical proof for the GM component merging procedure.

Our proposed HGI-PHD method can be applied to the space situational awareness region. It was shown that our proposed HGI-PHD filter outperformed the GM-PHD filter, the Am-GM-PHD filter, and the Do-GM-PHD filter, especially in scenarios where the detection probabilities are low, clutter rates are high, or there are closely spaced targets. the robustness and effectiveness of the proposed HGI-PHD filter were validated through simulation experiments and using real trajectories. Future work includes the development of efficient real-time algorithms for SAA tasks and the improvement of tracking accuracy for space targets.

**Author Contributions:** S.Z. proposed the conceptualization and methodology; Q.Y. wrote the draft manuscript; Z.W. and L.J. supervised the experimental analysis and revised the manuscript; Y.Z. proofread and revised the first draft. All authors have read and agreed to the published version of the manuscript.

**Funding:** This research received no external funding.

**Data Availability Statement:** Not applicable.

**Acknowledgments:** The authors would like to thank the editors and all the reviewers for their very valuable and insightful comments during the revision of this work.

**Conflicts of Interest:** The authors declare no conflict of interest.

## Appendix A

*Derivation of CRLB of the GTD Model*

Here, we derive the deterministic Cramer–Rao lower bound (CRLB) of the GTD scattering center model parameters. For simplicity, we give the CRLB derivation of the one-dimensional GTD model, and the CRB for the 3D-GTD model can be obtained by expansion. The one-dimensional GTD model can be written as

$$
\begin{aligned}
E(f_m) &= \sum_{i=1}^{I} A_i \exp(-\mathrm{j}4\pi f_0 r_i/\mathrm{c})(1 + \tfrac{m\Delta f}{f_0})^{\alpha_i} \exp(-\mathrm{j}4\pi m\Delta f r_i/\mathrm{c}) + \omega(f_m) \\
&= \sum_{i=1}^{I} a_i(1 + \tfrac{m\Delta f}{f_0})^{\alpha_i} \exp(-\mathrm{j}m\varpi_i) + \omega(f_m)
\end{aligned}
\tag{A1}
$$

where $\{r_i, a_i, A_i\}$ represent the position parameter, scattering intensity, and scattering type of the *i*-th scattering center, respectively. And other parameters have been defined in (6). Note that $a_i$ is equal to $A_i \cdot \exp(-\mathrm{j}4\pi f_0 r_i/\mathrm{c}) = a_{R_i} + \mathrm{j}a_{I_i} = |a_i|e^{\mathrm{j}\varphi_i}$, and $\varpi_i$ is equal to $-4\pi\Delta f r_i/\mathrm{c}$.

Thereafter, the estimated scattering parameters in (A1) can be expressed as follows:

$$
\zeta = \begin{bmatrix} \sigma^2 & \zeta_{a\mathrm{Re}}^{\mathrm{T}} & \zeta_{a\mathrm{Im}}^{\mathrm{T}} & \zeta_P^{\mathrm{T}} & \zeta_{\varpi}^{\mathrm{T}} \end{bmatrix}^{\mathrm{T}}
\tag{A2}
$$

where $\sigma^2$ denotes variance of Gaussian white noise, $\zeta_{a\mathrm{Re}}^{\mathrm{T}} = [a_{\mathrm{Re}1}, \ldots, a_{\mathrm{Re}I}]$, $\zeta_{a\mathrm{Im}}^{\mathrm{T}} = [a_{\mathrm{Im}1}, \ldots, a_{\mathrm{Im}I}]$, $\zeta_P^{\mathrm{T}} = [p_1, \ldots, p_I]$, and $\zeta_{\varpi}^{\mathrm{T}} = [\varpi_1, \ldots, \varpi_I]$.

When the relative operating bandwidth of radar satisfies $\gamma = \frac{N\Delta f}{f_0} = \frac{B}{f_0} \ll 1$, then we can obtain the following approximation:

$$
\left(1 + m\frac{\Delta f}{f_0}\right)^{\alpha_i} = \exp\left(\alpha_i \cdot \ln(m\frac{\Delta f}{f_0})\right) \approx \exp\left(\alpha_i \cdot m\frac{\Delta f}{f_0}\right)
\tag{A3}
$$

So, the 1D-GTD model is transformed as the damped exponential (DE) model, which is shown as

$$
E_{DE}(f_m) = \sum_{i=1}^{I} A_i \exp(m\alpha_i \Delta f/f_0)\exp(-4\pi\mathrm{j}f_m r_i/\mathrm{c}) + \omega(f_m) = \sum_{i=1}^{I} A_i v_i^m + \omega(f_m)
\tag{A4}
$$

where $v_i^m = p_i \cdot \exp(-4\pi\mathrm{j}f_m r_i/\mathrm{c})$ and $p_i = \exp(m\alpha_i \Delta f/f_0)$.

In [29], it has been verified that the CRLB of the DE model and the GTD scattering center model are substitutable by both theoretical derivations and simulation experiments. Therefore, the CRLB of the DE model is derived here to substitute that of the GTD model.

Due to being the Gaussian white noise, the CRLB matrix of the DE model can be computed as

$$
\mathbf{CRLB}_{DE} = \begin{bmatrix} \frac{\sigma^4}{N} & \mathbf{0}_{1\times 4I} \\ \mathbf{0}_{4I\times 1} & \frac{\sigma^2}{2}\mathbf{F} \end{bmatrix}
\tag{A5}
$$

$$F = \begin{bmatrix} \mathrm{Re}\left\{E + EB\Delta_1^{-1}BE\right\} & -\mathrm{Im}\left\{E + EB\Delta_1^{-1}BE\right\} & -\mathrm{Re}\left\{EBA\Delta^{-1}P\right\} & \mathrm{Im}\left\{EBA\Delta^{-1}\right\} \\ \mathrm{Im}\left\{E + EB\Delta_1^{-1}BE\right\} & \mathrm{Re}\left\{E + EB\Delta_1^{-1}BE\right\} & -\mathrm{Im}\left\{EBA\Delta^{-1}P\right\} & -\mathrm{Re}\left\{EBA\Delta^{-1}\right\} \\ -\mathrm{Re}\left\{P\Delta^{-1}A^{\mathrm{H}}BE\right\} & \mathrm{Im}\left\{P\Delta^{-1}A^{\mathrm{H}}BE\right\} & \mathrm{Re}\left\{P\Delta^{-1}P\right\} & -\mathrm{Im}\left\{P\Delta^{-1}\right\} \\ -\mathrm{Im}\left\{\Delta^{-1}A^{\mathrm{H}}BE\right\} & -\mathrm{Re}\left\{\Delta^{-1}A^{\mathrm{H}}BE\right\} & \mathrm{Im}\left\{\Delta^{-1}P\right\} & \mathrm{Re}\left\{\Delta^{-1}\right\} \end{bmatrix} \tag{A6}$$

where $0$ denotes the zero matrix, $\Delta = A^{\mathrm{H}}\Delta_1 A$, $A = \mathrm{diag}(a_1, a_2, \ldots, a_I)$, $\Delta_1 = B_2 - BEB$, $B_2 = V^{\mathrm{H}}N^2 V$, $B = V^{\mathrm{H}}NV$, $V = [v_1, v_2, \ldots, v_I]$, $v_i = v_i^l \cdot [1, v_i, \ldots, v_i^{N-1}]^{\mathrm{T}}$, $l = \begin{cases} -(N-1)/2, & N \text{ is odd} \\ -N/2 + 1, & N \text{ is even} \end{cases}$, $N = \mathrm{diag}(-(N-1)/2, \ldots, (N-1)/2)$, and $P = \mathrm{diag}(p_1, p_2, \ldots, p_I)$.

According to (A5) and (A6), it can be observed that it is complicated to obtain the Cramer–Rao lower bound of the DE model. However, for most wide-band radars, we have $\Delta f / f_0 \ll 1$, which is equivalent to $p_i \approx 1$. Therefore, by simplifying the CRLB matrix in (A5), the CRLB of the DE model can be obtained as

$$CRLB_{a_i} \approx \frac{\sigma^2}{2N} \tag{A7}$$

$$CRLB_{v_i} \approx \frac{6\sigma^2}{|a_i|^2 N^3} \tag{A8}$$

$$CRLB_{p_i} \approx \frac{6\sigma^2 p_i^2}{|a_i|^2 N^3} \tag{A9}$$

Based on (A4), the exact relations between the DE model parameters and the GTD model parameters are given by

$$r_i = -\frac{c}{4\pi\Delta f} \cdot \omega_i \tag{A10}$$

$$\alpha_i = \frac{f_0}{\Delta f} \cdot \ln p_i \tag{A11}$$

$$A_i = a_i \cdot \exp(\mathrm{j} 4\pi f_0 r_i / c) \tag{A12}$$

According to Equations (A7)–(A12), we obtain

$$\mathrm{var}\{r_i\} \geq \left(\frac{c}{4\pi\Delta f}\right)^2 \cdot \frac{6\sigma^2}{|a_i|^2 N^3} = \frac{3}{2\pi^2} \cdot \frac{1}{SNR_i} \tag{A13}$$

$$\mathrm{var}\{\alpha_i\} \geq \left(\frac{f_0}{\Delta f}\right)^2 \cdot \frac{6\sigma^2 p_i^2}{|a_i|^2 N^3} = \frac{6}{\gamma^2 \cdot SNR_i} \tag{A14}$$

$$\mathrm{var}\{|A_i|\} \geq \frac{\sigma^2}{2N} \tag{A15}$$

where $SNR_i$ denotes the peak signal-to-noise ratio of the $i$-th scattering center and $SNR_i \approx \frac{N|a_i|^2}{\sigma^2}$.

Finally, by extending the 1D-GTD model to the 3D-GTD model, we have

$$\mathrm{var}\{\hat{x}_i\} = \mathrm{var}\{\hat{y}_i\} = \mathrm{var}\{\hat{z}_i\} \geq \frac{3}{2\pi^2 SNR_{ii}} \tag{A16}$$

$$\mathrm{var}\{\hat{\alpha}_i\} \geq \frac{6}{\gamma^2 SNR_{ii}} \tag{A17}$$

$$\text{var}\{\hat{A}_i\} \geq \frac{1}{2SNR_{ii}} \tag{A18}$$

$$SNR_{ii} \approx \frac{MNK|A_i|^2}{\sigma^2} \tag{A19}$$

where $\sigma^2$ denotes the white Gaussian noise variance.

## Appendix B

Pseudo-code for the HGI-PHD filter

**given** $\left\{\overline{w}_{novel,k-1}^i, m_{k-1}^i, P_{k-1}^i\right\}_{i=1}^{M_{k|k-1}}$, the measurement set $\mathbf{Z}_k$

**step 1** Prediction of the HGI-PHD filter

$i = 0$.

for $m = 1, \ldots, M_{k-1}$

$i = i + 1$.

$\overline{w}_{novel,k|k-1}^i = p_{s,k} \overline{w}_{novel,k-1}^m$,

$m_{k-1}^m = F_{k-1} m_{k|k-1}^m$, $S_k^m = R_k + H_{k|k-1} P_{k|k-1}^m H_k^{\mathrm{T}}$.

$K_k^m = P_{k|k-1}^m H_k^{\mathrm{T}} \left[S_k^m\right]^{-1}$, $P_{k|k}^m = \left[I - K_k^m H_k\right] P_{k|k-1}^m$.

end

$M_{k|k-1} = i$.

**step 2** Construction of the HGI-PHD update components

for $m = 1, \ldots, M_{k|k-1}$

$\eta_{k|k-1}^m = H_k m_{k|k-1}^m$, $S_k^m = R_k + H_k P_{k|k-1}^m H_k^{\mathrm{T}}$,

$K_k^m = P_{k|k-1}^m H_k^{\mathrm{T}} \left[S_k^m\right]^{-1}$, $P_{k|k}^m = \left[I - K_k^m H_k\right] P_{k|k-1}^m$.

end

**step 3** Update of the HGI-PHD filter

for $m = 1, \ldots, M_{k|k-1}$

$\overline{w}_{novel,k}^m = (1 - p_{D,k}) \overline{w}_{novel,k|k-1}^m$,

$m_k^m = m_{k|k-1}^m$, $P_k^m = P_{k|k-1}^m$.

end

$n = 0$.

for each measurement $z \in \mathbf{Z}_k$

$n = n + 1$.

For $m = 1, \ldots, M_{k|k-1}$

$w_{novel,k}^{nM_{k|k-1}+m} = \dfrac{p_{D,k}(x_k) g_{hrrp}(hrrp_k) w_{k|k-1}^n \mathbf{q}_k^n(\mathbf{Z}_k)}{\lambda_c c_{hrrp}(hrrp_k) c_d(d_k) + p_{D,k}(x_k) g_{hrrp}(hrrp_k) \sum\limits_{i=1}^{N_{k|k-1}} w_{k|k-1}^n \mathbf{q}_k^n(\mathbf{Z}_k)}$

$m_k^{nM_{k|k-1}+m} = m_{k|k-1}^m + K_k^m(z - \eta_{k|k-1}^m)$,

$P_k^{jM_{k|k-1}+m} = P_{k|k}^m$.

end

$m_{\max} = \underset{m \in I}{\operatorname{argmax}}(w_k^i), m \in \left[1, nM_{k|k-1} + M_{k|k-1}\right]$, $\Re = \kappa_k(\lambda_c) + w_{novel,k}^{m_{\max}}\beta + \sum\limits_{i=1}^{N_{k|k-1}} w_{novel,k}^{jM_{k|k-1}+m}$,

$\overline{w}_{novel,k}^{nM_{k|k-1}+m} = \begin{cases} \dfrac{w_{novel,k}^{nM_{k|k-1}+m}}{\Re}, m \neq m_{\max}, m \in [1, M_k], n \in [1, N_{k|k-1}] \\ \dfrac{w_{novel,k}^{nM_{k|k-1}+m}\beta}{\Re}, m = m_{\max}, n \in [1, N_{k|k-1}] \end{cases}$.

end

$nM_{k|k-1} + M_{k|k-1}$.

**output** $\left\{\overline{w}_{novel,k}^i, m_k^i, P_k^i\right\}_{i=1}^{M_{k|k}}$.

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
