# Peer review of "Space Target Tracking with the HRRP Characteristic-Aided Filter via Space-Based Radar"

_remotesensing, doi:10.3390/rs15194808_

Round 1
Reviewer 1 Report (Previous Reviewer 1)
The authors have answered most comments in the revised manuscript. I have the last one question before publication.
When the HRPP information is used, it means the wide-band signal is used. In this case, narrow signal is used to track, how do these two signals work simultaneously in the radar system? Please give the explanation.
I have no comments on the quality of English language.
Author Response
Please see the attachment.

Reviewer 2 Report (Previous Reviewer 2)
This manuscript proposes an improved PHD algorithm for multiple space target tracking (MTT) for space situational awareness (SSA). The information of high-resolution range profile (HRRP) of the targets is used to aid the filtering algorithm. The simulation result demonstrates the effectiveness of the proposed framework.
This manuscript presents the very details of the flow and holds good technical soundness. However, some improvements have to be made before publication.
1. Ensure the consistency between Fig. 2 and the main text. For example, put the section numbers in Fig. 2 so that the connection can be easily found. The procedures in these subsections also need to be depicted in Fig. 2 appropriately if necessary.
2. In 4.1, the setup uses 4 scattering centers. Why not use those in 4.2?
3. HRRP prediction is an important step. An evaluation of its performance is needed.
4. The writings should be thoroughly and carefully revised. It is strongly suggested to rewrite the ‘abstract’ and ‘introduction’ parts.
A few examples:
4.1 Abstract
(1) ‘in (SSA) region’ or in (SSA) field’?
(2) ‘the space-based radar system is implemented in this paper’. Implements means to put (a decision, plan, agreement, etc.) into effect, or to realize. ‘Implemented’ is not suitable here.
(3) ‘many potential valuable messages are missed’. What does ‘messages’ refer to?
(4) Some sentences (e.g. ‘Finally,…’) are too long to understand.
4.2 Introduction
(1) ‘it processes a stronger early warning ability for high speed space targets.’ Process usually means to perform a series of operations on something. This sentence is quite confusing.
(2) Again, please use ‘implement’ appropriately.
(3) ‘the SSA surveillance tasks conclude the effective’. What does ‘conclude’ refer to?
(4) ‘effective discrimination problems will occur, and they are challenging to be solved’. This problem is not considered in the following texts.
(5) ‘these recent years’ can be simplified to ‘recent years’.
(6) section 1.2 should be carefully reconsidered. For Problem 1, HRRP has been used for tracking ground targets. Problem 2 is not handled in this manuscript. Problem 4 is not on the same level as the first three problems.
4.3 Other parts
(1) Please ensure that EVERY symbol is explained in the text.
(2) The title of section 3 can be ‘Propose method.’
(3) The title of section 3.1 can be ‘Parameter Estimation of Scattering Centers’
(4) ‘Hermitten’ should be ‘Hermitian’
(5) The title of section 5 can be simply ‘Simulation results.’
(6) The title of section 5.1 can be ‘Performance evaluation of scattering center estimation.’
English should be carefully improved.
Author Response
Please see the attachment.

Reviewer 3 Report (New Reviewer)
I am not an expert in radars but I have mainly two remarks / request for modifications:
- intro can be improved to present better the need for related work (and your work). It is always a compromise between what you present and what you assume the reader knows, and I think you went very deep, very fast. See the excellent introduction made here: https://arc.aiaa.org/doi/10.2514/1.G002067
- the statement you provided for 4.6, and the conclusions might be wrong. I do not believe you really ran Matlab 2021 on Intel Core processor (first generation, year 2006?). Please state exactly what processor you used, not just 3.4 GHz Intel Core (was it i5-7500, i7-6700 ?). Also say specifically how many cores you have used out of the available and what OS (for example, in linux you can list cpu usage every X ms [see https://www.baeldung.com/linux/process-periodic-cpu-usage], and then average that to have a proper estimation of cpu usage throughout the run of the algo). Also, run with multiple clutters (x1,x2,x4,x8,x16...) to see how it scales with power-of-two sizes. You only ran on 50 and 100: why have you chosen these values, and why only two of them? Also specify exactly what you measured: I have often seen data generation consuming a lot of the runtime, hence hiding the algo complexity. Generate the data and store it into ram, start the timer, run the algo on the previously RAM-stored data, then stop the timer. Have you made an analysis on the complexity of the algos (O(1), O(N)., O(NlogN)) ? Was this confirmed by the experiments?
Round 2
Reviewer 1 Report (Previous Reviewer 1)
The authors have answered my questions after two rounds of iteration and modification. The quality of the paper has been greatly improved, and it is now acceptable for publication.
Reviewer 2 Report (Previous Reviewer 2)
I have no further comments.
I understand that the authors tried hard to improve the English. Although the current version is more readable, I still recommend that they keep polishing the writing.
This manuscript is a resubmission of an earlier submission. The following is a list of the peer review reports and author responses from that submission.
Round 1
Reviewer 1 Report
This paper proposes an adapting PHD-based filter with the aided HRRP characteristics and an adapting threshold for GM components merging procedure to tackle the space targets false tracking problems. This work is interesting. However, the following issues need to be improved:
(1) In the abstract, when the abbreviation “PHD” appears for the first time, it needs to give the full name, please check.
(2) When the HRPP information is used, it means the wide-band signal is used. In this case, narrow signal is used to track, how do these two signals work simultaneously in the radar system? Please give the explanation.
(3) In Table 2, the radar parameters are suggested to be given for ease of understanding.
(4) The author's name in the References, either abbreviated or fully spelled, it should be consistent and conform to the journal style.
This paper proposes an adapting PHD-based filter with the aided HRRP characteristics and an adapting threshold for GM components merging procedure to tackle the space targets false tracking problems. This work is interesting. However, the following issues need to be improved:
(1) In the abstract, when the abbreviation “PHD” appears for the first time, it needs to give the full name, please check.
(2) When the HRPP information is used, it means the wide-band signal is used. In this case, narrow signal is used to track, how do these two signals work simultaneously in the radar system? Please give the explanation.
(3) In Table 2, the radar parameters are suggested to be given for ease of understanding.
(4) The author's name in the References, either abbreviated or fully spelled, it should be consistent and conform to the journal style.
Reviewer 2 Report
This manuscript deals with multiple space target tracking using space-borne radar. The PHD algorithm is improved by incorporating HRRP information. To predict the HRRP, an improved ESPRIT algorithm is proposed to estimate the scattering centers based on the GTD model, and a likelihood is proposed for the PHD filtering as well. Simulation result demonstrates the effectiveness of the proposed framework.
Major concerns:
1. The manuscript emphasize the ‘clutter’ several times. What are the clutters in the space environment? Also, please analyze the influences of clutter in simulations.
2. The proposed 3D-ESPRIT aims to estimate the scattering centers in low SNR conditions. Please explain the theoretical foundations and the differences to Ref [27]. In simulation 4.1, please show the HRRPs and the estimated scatter centers. Why choose Ref [15] as comparison. Also, please analyze the influences of distribution of scattering centers.
3. Please detail the concept and derivation of the proposed the HRRP likelihood. For example, what is the PDF of HRRP?
4. It seems that the characteristic of HRRP used in the proposed PHD framework is the correlation (or similarity) between adjacent times. It is known that HRRP is sensitive to the aspect. Please explain the corresponding influences. Also, please analyze the HRRP prediction performances, and provide the trajectory results.
5. The organizations and the writings require significant effort to be improved.
Here are just some examples:
(1) Page 1, what is the role of MTT in SSA? Why ‘the SSA tasks can be regarded as multiple target tracking (MTT) problems’?
(2) Page 2, the literature review is hard to get the main point.
(3) Page 3, there are lots of inappropriate conjunctions: ‘It should be mentioned’, ‘Next’, ‘Note that’.
(4) Page 3, the LMB, GLMB lack of references.
(5) Page 3, para. 4, the background of space radar should be moved to beginning of introduction.
(6) Page 3, what are the aims of stating the drawbacks 1-4.
(7) Page 3, 1.2, there are no ‘problem analysis’.
(8) Page 4, please give the definitions of the symbols, such as v_k, delta_w. Please pay attention to other equations hereafter.
(9) Page 5, in GTD, there is no delta_f.
(10) Page 5, what is P_xi, P_yi, P_zi in eq. 7-10.
(11) Page 8, the title of section 3 is too long. ‘Our’ in the title of 3.1 can be removed.
(12) Page 10, what is the difference of zk in eq. 38 and zk in eq. 43?
(13) Page 13, line 411, the eq. 46 should not appear in the main text.